# Validity and Reliability of a Bilingual Healthcare Discrimination Scale Among Churchgoing Latino Adults in Los Angeles

**DOI:** 10.3390/bs15111514

**Published:** 2025-11-07

**Authors:** Daniel F. López-Cevallos, Mariana Pinto-Alvarez, Karen R. Flórez, Kathryn P. Derose

**Affiliations:** 1Department of Health Promotion and Policy, School of Public Health and Health Sciences, University of Massachusetts Amherst, Amherst, MA 01003, USA; mpintoalvare@umass.edu (M.P.-A.); kpderose@umass.edu (K.P.D.); 2Center for Systems and Community Design, Graduate School of Public Health and Heath Policy, City University of New York, New York, NY 10027, USA; karen.florez@sph.cuny.edu; 3Department of Behavioral and Policy Sciences, RAND, Santa Monica, CA 90401, USA

**Keywords:** healthcare discrimination, psychometrics, health disparities, race/ethnicity, Latinos

## Abstract

Healthcare discrimination is an important barrier to accessing services among Latino populations in the United States. However, few validated scales have been developed to systematically examine this issue. In this study, we evaluated the validity and reliability of a bilingual healthcare discrimination scale in a sample of churchgoing Latino adults in Los Angeles, California. The study sample included 336 participants (foreign-born: 250; US-born: 86) who attended 12 Catholic churches in Los Angeles. Psychometric testing of the 7-item healthcare discrimination (HCD) scale included internal consistency; split-half reliability; convergent, discriminant, and predictive validity; and confirmatory factor analyses. The HCD had relatively high internal consistency (full sample Cronbach’s α = 0.92; foreign-born: 0.91; US-born: 0.92) and showed good convergent and discriminant validity, as it was moderately correlated with the depression scale (full sample r = 0.28, *p* < 0.001) and weakly correlated with the acculturation scale (full sample r = 0.15, *p* = 0.008). Confirmatory factor analyses yielded further support for a one-factor solution. Our study finds that the HCD is a valid and reliable scale for use among churchgoing Latino adult populations in the United States. Future studies should examine the psychometric properties of the HCD among Latinos of diverse backgrounds, geographic locations, religious beliefs, and languages.

## 1. Introduction

Racial/ethnic discrimination and structural racism are critical public health issues in the United States (US) ([3]; [60]). Discrimination, the process of making unfair or prejudicial distinctions between people based on groups, classes, or other categories to which they belong or are perceived to belong (e.g., race/ethnicity, gender, or sexual orientation), is an expression of broader structural, systemic, and institutional biases ([60]). In the United States, racism and discrimination can be traced back to the inception of the nation-state, creating a false racial hierarchy that upholds racial/ethnic inequities to this day ([20]; [21]; [42]).

In healthcare settings, racial/ethnic discrimination occurs when patients receive unequal treatment (based on their race/ethnicity) as a result of biased attitudes, stereotyping, or systemic barriers/policies that lead to disparities in access, quality, and outcomes ([40]; [55]). Racial/ethnic discrimination is a significant barrier to access and utilization of healthcare among minoritized populations ([30]; [52]; [54]). For instance, a recent survey found that one in five adults experienced discrimination in the healthcare system, with racial/ethnic discrimination being the most common (17%) ([43]). In addition, a contemporary study using a nationally representative sample of US Latina/o and non-Latina/o White adults found that 20% of Latinos reported experiencing discrimination when going to a doctor or health clinic (compared to 5% among non-Latina/o White adults) ([17]). Earlier prevalence estimates vary depending on the study sample, with studies reporting race/ethnicity-based discrimination ranging from 7% to 52% for African Americans, and 4% to 25% for Latinos ([6]; [17]; [24]; [31]; [46]; [56]).

Racial/ethnic minoritized groups experience disproportionally higher rates of illness and death across a wide range of health conditions, compared to their non-Latino white counterparts ([25]). The COVID-19 pandemic further confirmed these long-standing inequities ([23]). The complex, expensive, and fragmented landscape of the US healthcare system makes it even more difficult for racial/ethnic minorities and low-income populations to access quality care ([38]; [65]). Studies have found that racial/ethnic discrimination in healthcare settings can lead to poorer outcomes, including new or worsened disability among older adults ([48]), depressive symptoms ([61]), reduced use of cancer screening services ([22]), poorer quality of care ([47]; [56]), lower satisfaction with healthcare services ([37]), and lower Consumer Assessments of Healthcare Providers and Systems (CAHPS) scores ([63]).

Despite a growing body of research, studies tend to rely on questions that have not been theoretically developed or psychometrically validated ([53]), and there are few instruments for accurately measuring healthcare discrimination among Latinos, the largest racial/ethnic minority group in the US. Indeed, much of this research relies on a question or sets of questions that have not necessarily been theoretically developed and/or validated ([52]; [53]). A review of the evidence found that most studies used a single item on healthcare discrimination from the widely used *Experiences of Discrimination* Scale ([17]; [27]; [43]).

Building on questions first developed by Bird & Bogart ([7]), the *Discrimination in Medical Settings Scale* ([45]) adapted an everyday discrimination scale ([12]; [64]) to medical settings. Their exploratory factor analysis was conducted with a relatively small sample (n = 74) of older (mean age = 66 years) African Americans in Chicago ([45]). A 2022 review found that while researchers have used the scale without making major textual modifications, few studies have gone beyond testing internal consistency (Cronbach’s alpha) ([58]). In a 2019 study ([35]), researchers first translated and then validated a bilingual (Spanish/English) version of the Peek scale among young adult Latinos (ages 18–25), mostly of Mexican-American background, and living in rural Oregon. A recent study using this bilingual healthcare discrimination (HCD) scale found that lower income Latinos had significantly higher perceived healthcare discrimination, compared to higher income non-Latina/o White individuals ([51]).

While these two studies ([35]; [51]) provide promising evidence, it remains unclear whether the scale performs similarly well among other Latino groups (e.g., older, living in urban areas, churchgoers). According to the Census Bureau, Latinos comprise over 19% of the US population (64 million people) ([59]). There is evidence that the Latino population has dispersed widely since at least the 1990s, moving away from historically established communities in the US Southwest for fast-growing “new destinations” in urban and rural communities across the country ([32]). Currently, there are 13 US states with one million or more Latino residents. In each of these states, a relatively large share of the US Latina/o population are still concentrated in major metropolitan areas. For instance, Latinos make up 48% (1.8 million people) of the population in Los Angeles, CA ([10]), and 28% (2.4 million) of the population in New York, NY ([11]).

Moreover, studying this issue among Latino churchgoers is important, as 70% of Latino adults identified themselves as religious ([29]), and religious involvement has been shown to buffer the impact of discrimination on health outcomes ([16]). Furthermore, religious institutions may provide critical protection to Latinos, either when they are initially exposed to discrimination or when they endure persistently high levels of discrimination (e.g., when accessing services or in the workplace). Indeed, research suggests that religious social capital can attenuate the impact of immigration-related stress among Latinos ([49]). To address these gaps, we examined the psychometric properties of the bilingual HCD scale in a sample of churchgoing (US- and foreign-born) older Latino adults in Los Angeles, California.

## 2. Materials and Methods

### 2.1. Study Participants

De-identified secondary data were obtained from a Los Angeles-based parent study, which was a cluster randomized controlled trial intervention linking predominantly Latino Catholic churches with their local parks to increase physical activity among Latino parishioners ([15]). For implementation purposes, the study was conducted in two cohorts of churches. The bilingual healthcare discrimination scale (HCD) ([35]) was included in the follow-up questionnaires for both the first and second cohorts (fielded in 2022–2023, and 2024–2025, respectively). The research protocol was approved by RAND’s Human Subjects Protection Committee. Each participant provided written informed consent before enrolling in the study. The final sample, combining both cohorts, included 336 participants with no missing data for any of the study variables.

### 2.2. Measures

The *Health Care Discrimination (HCD) scale* included seven prompts to the question “When getting healthcare of any kind, have you ever had any of the following things happen to you because of your race or ethnicity?” A mean score of the seven items rated on a 5-point scale (ranging from never to always) was computed, with higher scores indicating greater discrimination (Cronbach’s α = 0.92). While limited to questions fielded by the parent study, selection of relevant covariates was guided by two complementary theoretical frameworks: intersectionality theory, and the minority stress model. Intersectionality theory ([8]; [62]) describes how holding multiple stigmatized identities (e.g., racial/ethnic/gender minorities; low socioeconomic status) can lead to increased experiences of discrimination and poorer healthcare outcomes, while the minority stress model ([18]; [41]) describes how bias and discrimination can affect excess stress for minority populations. Hence, relevant covariates included the following: (1) The *Patient Health Questionnaire depression scale (PHQ-8)* includes eight items assessing diagnostic and severity measures for depressive disorders ([28]). Responses ranged from not at all to nearly every day, with points (0–3) assigned to each category. The scores for each item were then summed to produce a total score between 0 and 24. A mean score was computed, with higher scores indicating greater depression (Cronbach’s α = 0.85); (2) The *Brief Acculturation Scale for Hispanics* includes four items from Marin et al.’s Language Use subscale ([39]; [44]). The language items were rated on a 5-point scale, ranging from only Spanish to only English. A mean score was computed, with higher scores indicating greater acculturation (Cronbach’s α = 0.95); (3) The *Perceived Stress Scale* includes four items asking about the respondents feelings and thoughts in the previous month ([13]). Responses ranged from never to very often, on a 5-point scale. Scores were then summed, ranging from 0 to 16, with higher scores indicating more stress (Cronbach’s α = 0.50).

Sociodemographic covariates included nativity (Foreign; US-born) and religious attendance (over the past four weeks, how many worship services or activities did you attend at this church? Options ranged from none to four or more), age, sex (female; male), educational level (seven categories, ranging from 6th grade or less to some graduate school or graduate degree), marital status (six categories, ranging from single/never married to widowed), health insurance status (uninsured vs. insured), annual household income (nine categories, ranging from $9999 or less to $100,000 or more), and household size (number of adults and children).

### 2.3. Statistical Analyses

A correlation matrix of all study variables is shown on Table 1. Following similar procedures as the previous study ([35]), we conducted psychometric testing for the HCD scale overall, and by nativity (Foreign or US-born). Reliability was evaluated using Cronbach’s alpha for internal consistency (calculated for the PHQ-8 and BASH scales) and split-half reliability analysis. We then conducted confirmatory factor analyses (CFA) to further examine the one-factor structure of HCD in this church-based sample. CFA was chosen because previous HCD studies have identified a one-factor structure ([35]; [45]). Hence, we hypothesized that a single latent factor would explain the relationships among the seven HCD items ([9]). Loadings of 0.4 or higher were considered acceptable; higher loadings indicated a stronger connection between the observed variable and the latent factor. Following standard practices ([4]; [9]), four fit indices guided our determination of model fit: (1) the root mean square error of approximation (RMSEA–indicates the degree of model misspecification; values of 0.05 or below indicate good fit, while values up to 0.08 may be acceptable), (2) the standardized root mean square residual (SRMR–describes how far the model-implied correlation differs from the actual correlations among the observed variables; it is recommended that the SRMR should be 0.10 or lower); (3) the robust comparative fit index (CFI–compares the fit of the model with the fit of the null model; values greater than 0.9 indicate good fit); and (4) 1) chi-square (*X*^2^–which tests whether the model-implied covariance matrix is different from the sample matrix). As a rule of thumb, models were considered acceptable when two or more indices indicated a good model fit ([9]; [26]).

In CFA, minimum sample sizes are recommended to limit the non-convergence probability of unbiased estimates. As a rule of thumb, the ratio of cases to free parameters is commonly used in CFA studies, with minimum recommendations ranging from 10:1 to 20:1 ratios ([50]). For our 7-item scale, we would need between 70 and 140 cases (participants). Hence, our study sample of 336 participants (a 48:1 ratio) is sufficiently powered to conduct a CFA analysis.

The construct validity of the HCD was assessed using Pearson’s correlations between the HCD and PHQ-8 scales (convergent validity) and between the HCD and BASH scales (discriminant validity). In principle, strong (and statistically significant) correlations would support convergent validity; in turn, weak (and non-significant) correlations would support discriminant validity ([57]). A linear regression was used to evaluate the predictive validity of the HCD on perceived stress. Stata SE 18.5 (College Station, TX, USA) was used for all statistical analysis. We followed the COSMIN reporting guidelines ([19]).

## 3. Results

Most respondents were female (75.9%), foreign-born (74.4%), had a high school-level education or lower (75.9%), had some form of health insurance (88.7%), had an income below $50,000 (73.5%), and responded to the survey in Spanish (83.3%). Their median age was 56 years (range: 21–90). Compared to US-born, foreign-born respondents had similar healthcare discrimination scores (0.35, sd = 0.65 vs. 0.44, sd = 0.75, *p* = 0.286) while depicting lower levels of depression (2.77, sd 3.61 vs. 3.97, sd = 4.80, *p* = 0.027), acculturation (1.46, sd = 0.68 vs. 3.21, sd = 1.27, *p* < 0.001), and stress (4.40, sd = 3.12 vs. 5.27, sd = 2.88, *p* = 0.025).

As shown on Table 1, the HCD scale had a relatively weak and negative correlation with age (r = −0.17, *p* = 0.002) and marital status (r = −0.11, *p* = 0.041); while weak and positive with educational level (r = 0.15, *p* = 0.008). The strongest (albeit still weak and positive) correlation was found with depressive symptoms (r = 0.28, *p* < 0.001). Confirmatory factor analyses (CFA, see Table 2) yielded further support for the one-factor solution of previous validation studies, depicting significant standardized factor loadings ranging from 0.63 to 0.84 and a relatively good model fit (RMSEA = 0.19; SRMR = 0.05; CFI = 0.90; *χ*^2^(14) = 186.13, *p* < 0.001). We found a similar pattern for both foreign- and US-born subsamples.

Table 3 shows that the HCD had relatively high internal consistency (full sample Cronbach’s α = 0.92; foreign-born: 0.91; US-born: 0.92). The individual reliability of all HCD items was >0.89. Split-half reliability was robust for the full sample (r = 0.89, *p* < 0.001) and by nativity (foreign-born r = 0.89, *p* < 0.001; US-born r = 0.91, *p* < 0.001). The mean HCD score was 0.37 (SD = 0.68). The HCD showed relatively good convergent and discriminant validity as it was moderately correlated with the PHQ-8 (full sample r = 0.28, *p* < 0.001; foreign-born r = 0.24, *p* < 0.001; US-born r = 0.35, *p* < 0.001) and weakly correlated with the BASH (full sample r = 0.15, *p* = 0.008; foreign-born r = 0.14, *p* = 0.031; US-born r = 0.16, *p* = 0.140). Regarding predictive validity, HCD was significantly associated with perceived stress in the unadjusted models for the full (*β* = 0.70, *p* = 0.005) and the foreign-born sample (*β* = 1.02, *p* = 0.001) but not for the US-born (*β* = −0.10, *p* = 0.816). After adjusting for the PHQ-8, BASH, and sociodemographic characteristics, HCD was no longer significantly associated with perceived stress.

## 4. Discussion

The present study is the first to test the bilingual (Spanish/English) healthcare discrimination (HCD) scale in a sample of churchgoing Latinos in Los Angeles, California. Since this bilingual HCD was first validated in 2019, no studies to date have examined its validity and reliability among other Latino subgroups. Hence, we extend the reach of the original validation study from 18-to-25-year-olds living in rural Oregon ([35]) to churchgoing older adults living in a major metropolitan area (Los Angeles), home to one of the largest Latino communities in the US and a population shaped by intersecting ethnic, religious, linguistic, and immigrant identities. Indeed, the mean values of each of the seven items were remarkably lower in the current sample than in the 2019 study with young adult Latinos. For instance, the mean of item # 1(“being treated with less courtesy than other people”) was 0.43 (*sd* = 0.84) among Latino churchgoers (compared to 0.85, *sd* = 0.98) among young adult Latinos). The marked difference may be due to several factors, such as location (urban Los Angeles vs. rural Oregon), age (older adults vs. young adults), religious affiliation (church-based sample vs. community sample), and years living in the US (75% of participants in the present study were foreign-born vs. 59% in the Oregon study). Nevertheless, factor loadings were very similar across the two studies, which may support the use of the HCD scale in future Latino-focused studies.

One of the strengths of our analysis is the use of two relevant theoretical frameworks, intersectionality theory and the minority stress model, to inform the inclusion of relevant constructs. Such an approach can deepen our understanding of how instances of discrimination in healthcare settings are experienced by minoritized populations. Moreover, it can offer tools for healthcare systems and providers to respond to systemic experiences of discrimination in and outside healthcare settings. Future research should leverage this scale to further analyze the complex, layered nature of discrimination, and therefore enhance our understanding of how systemic inequities in healthcare are experienced within this large and diverse group ([16]; [43]). For instance, multilevel, mixed-methods, and longitudinal designs can better capture how racial/ethnic discrimination operates at the interpersonal, institutional, and structural levels in healthcare settings. They should also test interventions and apply intersectional life-course frameworks to reveal how these layered forms of discrimination accumulate and affect health outcomes over time. Moreover, studies should consider supplementing the use of the HCD scale with other measures that can more robustly capture “unequal treatment” such as medical mistrust ([36]).

Over two decades ago, the seminal 2003 *Unequal Treatment* report pointed out how healthcare providers’ bias, prejudice, and discrimination contribute to racial/ethnic health disparities, even after accounting for other relevant factors (e.g., income levels, insurance status) ([55]). Despite the report’s high visibility, a recent review of the evidence ([40]) concluded that little progress has been made to date in addressing the negative effects of discrimination on health, as demonstrated by the paucity of studies examining the association between discrimination and access to and utilization of healthcare services among racial/ethnic minorities ([1]; [2]; [5]; [14]; [22]). Although discrimination in healthcare settings is relatively well documented in the literature, few valid and reliable scales accurately measure this phenomenon. The present study contributes to the knowledge base by extending the validity and reliability of the seven-item bilingual HCD scale for other Latino subgroups, which can be used in clinical, community, and population-based studies.

Our psychometric study had several limitations, including the fact that our analyses were limited in scope to the variables included in the parent study. For example, perceived stress may have been too distal an outcome to use for testing predictive validity. The parent study did not include more proximal outcomes, such as the number of healthcare visits in the previous year, which should be explored in future studies. Moreover, the sample was recruited from selected Catholic churches in Los Angeles, which may not reflect the experiences of other Catholic Latinos and certainly does not capture the experience of Latinos of other religious affiliations. This is a relevant area for future research, as previous work has documented differences in medical mistrust between Latinos attending Catholic versus Pentecostal churches ([33]). Third, the comparatively small sample size of US-born Latinos may have negatively influenced the CFA model fit (e.g., potentially leading to less stable parameter estimates or a lower statistical power to detect model misfit, which in turn can make the results less generalizable).

Fourth, we did not test the HCD in the same individuals twice to assess test–retest reliability. Fifth, since there was a time lapse between the two cohorts included in this sample, their experiences of healthcare discrimination may have differed between the two time periods (2021–2022 and 2023–2024). Sixth, while we closely followed the methodology of the previous validation study among young-adult Latinos in Oregon, we could not compare the psychometric properties of the HCD scale by language, as most of the participants (83%) in the parent study responded in Spanish. However, examining the HCD psychometric properties for foreign vs. US-born Latinos is relevant, as previous research has found differences in healthcare discrimination between these two groups ([34]).

## 5. Conclusions

The present validation study provides further evidence to support the use of the bilingual healthcare discrimination (HCD) scale among diverse Latino populations in the US in both clinical and population-based studies. Future research should examine the psychometric properties of the HCD among Latinos of different ethnic backgrounds, geographic locations, languages and religious beliefs. More generally, using validated scales to monitor progress (or lack thereof) towards addressing racial/ethnic healthcare discrimination in the U.S. is crucial for identifying inequities and guiding programmatic and policy changes to ensure that all communities receive fair, respectful, and effective healthcare services.

## Figures and Tables

**Table 1 behavsci-15-01514-t001:** Pearson’s correlations between HCD scale and other study variables (n = 336).

	HCD Scale	1	2	3	4	5	6	7	8	9	10	11	12
1. PHQ-8	0.28 ***	1											
2. BASH	0.15 **	0.15 **	1										
3. PSS	0.16 **	0.40 ***	0.09	1									
4. Nativity	0.06	0.12 *	0.66 ***	0.12 *	1								
5. Age	−0.17 **	−0.09	−0.36 ***	−0.11 *	−0.37 ***	1							
6. Sex	0.04	−0.05	0.16 **	0.02	0.08	−0.05	1						
7. Educational level	0.15 **	0.16 **	0.62 ***	0.10	0.43 ***	−0.43 ***	0.08	1					
8. Marital status	−0.11 *	−0.03	−0.16 **	0.05	−0.17 **	0.38 ***	−0.06	−0.19 ***	1				
9. HIS	0.03	−0.13 *	−0.02	0.00	−0.04	−0.17 **	0.08	0.00	−0.10	1			
10. Income	0.06	−0.01	0.41 ***	−0.07	0.23 ***	−0.28 ***	0.17 **	0.40 ***	−0.17 **	0.03	1		
11. HSA	0.05	0.04	0.02	−0.04	0.03	−0.25 ***	−0.05	0.02	−0.18 **	0.13 *	0.18 **	1	
12. HSC	0.09	−0.13 *	−0.04	−0.06	−0.07	−0.32 ***	−0.03	−0.01	−0.07	0.05	0.02	0.23 ***	1
13. Religious attendance	0.03	0.01	−0.03	−0.03	−0.00	0.30 ***	−0.04	−0.09	0.08	−0.10	0.03	−0.02	−0.12 *

Abbreviations: HCD (Healthcare Discrimination Scale); PHQ-8 (Personal Health Questionnaire-8 Depression Scale); BASH (Brief Acculturation Scale for Hispanics); PSS (Perceived Stress Scale); HIS (Health Insurance Status); HSA (Household Size–Adults); HSC (Household Size.–Children). * *p* < 0.05, ** *p* < 0.01, *** *p* < 0.001.

**Table 2 behavsci-15-01514-t002:** Summary Statistics and Factor Loadings for the Healthcare Discrimination Scale (HCD) items among churchgoing Latinos in Los Angeles (n = 336).

Item	Mean (SD)			Factor Loading		
	Full Sample	Foreign-Born	US-Born	Full Sample	Foreign-Born	US-Born
1. Been treated with less courtesy than other people.	0.43 (0.84)	0.42 (0.80)	0.49 (0.94)	0.76 *	0.76 *	0.74 *
2. Been treated with less respect than other people.	0.38 (0.76)	0.38 (0.78)	0.38 (0.72)	0.83 *	0.82 *	0.91 *
3. Received poorer service than others.	0.37 (0.70)	0.36 (0.69)	0.41 (0.73)	0.84 *	0.82 *	0.94 *
4. Had a doctor or nurse act as if he or she thinks you are not smart.	0.30 (0.70)	0.27 (0.66)	0.40 (0.83)	0.83 *	0.83 *	0.78 *
5. Had a doctor or nurse act as if he or she is afraid of you.	0.13 (0.55)	0.10 (0.46)	0.24 (0.75)	0.63 *	0.66 *	0.60 *
6. Had a doctor or nurse act as if he or she is better than you.	0.30 (0.71)	0.28 (0.68)	0.35 (0.79)	0.83 *	0.84 *	0.76 *
7. Felt like a doctor or nurse was not listening to what you were saying.	0.39 (0.77)	0.38 (0.77)	0.41 (0.76)	0.77 *	0.76 *	0.76 *
All 7 items	0.37 (0.38)	0.35 (0.65)	0.44 (0.75)			

Abbreviations: SD (Standard Deviation); * *p* < 0.001.

**Table 3 behavsci-15-01514-t003:** Internal Consistency (Cronbach’s alpha), and Pearson’s Correlations examining the convergent (PHQ-8) and discriminant (BASH) validity of the HCD, for the full sample of churchgoing Latinos in Los Angeles, and by nativity.

			*Pearson’s Correlations*	
*Scale*	*Mean (SD)*	*Cronbach’s alpha*	1	2
**Full sample (n = 336)**				
1. HCD	0.37 (0.68)	0.92	1.00	
2. PHQ-8	3.05 (3.97)	0.85	0.28 ***	1.00
3. BASH	1.90 (1.16)	0.95	0.14 **	0.15 **
**Foreign-born (n** **=** **250)**				
1. HCD	0.35 (0.65)	0.91	1.00	
2. PHQ-8	2.77 (3.61)	0.82	0.24 ***	1.00
3. BASH	1.46 (0.68)	0.91	0.14 *	0.04
**US-born (n = 86)**				
1. HCD	0.44 (0.75)	0.92	1.00	
2. PHQ-8	3.87 (4.80)	0.91	0.35 ***	1.00
3. BASH	3.21 (1.27)	0.92	0.16	0.15

Abbreviations: HCD (Healthcare Discrimination Scale); PHQ-8 (Personal Health Questionnaire-8 Depression Scale); BASH (Brief Acculturation Scale for Hispanics). * *p* < 0.05; ** *p* < 0.01; *** *p* < 0.001.

## Data Availability

The data in this study are available from the corresponding author upon reasonable request.

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
