# Peer review of "Validity and Reliability of a Bilingual Healthcare Discrimination Scale Among Churchgoing Latino Adults in Los Angeles"

_behavsci, 2025, doi:10.3390/bs15111514_

Round 1

Reviewer 1 Report

Comments and Suggestions for Authors

see attached

Author Response

Response to reviewers

Reviewer # 1

Summary: This study supports the validity and reliability of a well known scale, the Discrimination in Healthcare Scale (Peek, 2011) in an adult population of Latinos in Los Angeles using a similar methodology as a prior study by the same author focusing in Latino young adults. Healthcare discrimination interferes with care and healthcare outcomes, including among Latinos. The study adds to the access to healthcare and healthcare literature as it demonstrates this scale is suitable to evaluate discrimination as a sole construct among Latinos adults living in the United States. Having a bilingual and short scale assessing discrimination can complement other surveys addressing Latinos healthcare disparities. I have some questions and suggestions on how to improve the manuscript with only minor revisions.

Introduction: Please, provide a brief definition for healthcare discrimination and support it with literature on how Latinos may perceive it in hospital settings and in other spaces. This will support the need to continue to consider discrimination as a potential factor in healthcare and health outcomes.

AR: Thank you. We have now provided a brief definition of healthcare discrimination and included relevant citations regarding Latino population experiences when accessing healthcare services (pages 1-2).

I was also wondering how the scale has been used and found a great systematic review. Use it to support a knowledge gap in assessing psychometric properties of discrimination in healthcare scales. See if there is any other relevant publication that you could use to compare and contrast your findings.

Thorburn, S., & Lindly, O. J. (2022). A systematic search and review of the discrimination in health care measure, and its adaptations. Patient education and counseling, 105(7), 1703-1713.

AR:  Thank you. We have incorporated this citation into the Introduction (page 3) and Discussion (page 8) sections to compare and contrast our findings.

Add a reference line 42 for “disproportionate rate of illness” among racial minorities.

AR: Thank you. We have added a reference (Howard et al., 2014) to this statement.

Methodology:

  1. Religious attendance is not reported in the correlation table.

AR: Thank you. We have now included religious attendance in the correlation table (page 5).

  1. Nativity or country of origin is not written in the sociodemographics (line 130).

AR: Thank you. We now mention nativity (foreign/US-born) among the sociodemographic variables included in the study (consistently across the manuscript).

  1. Add a brief explanation on how you will be interpreting factor loading, why you chose this approach, predetermined cutoffs for each model fit, and provide proper references.

AR: We have added a brief explanation of the interpretation of factor loadings, the reason for choosing CFA (e.g., building on a previous study that confirmed a one-factor structure), and cutoffs for model fit indices with relevant citations (page 5).

Results:

  1. Correlation
  • Is it country of origin the same as nativity (foreign or U.S. born)? If so, please clarify and be consistent across the manuscript. (Line 177) Also, if a variable was dichotomized, please add the reference category (e.g., single vs. married)

AR: Thank you. We are now consistently using nativity (foreign vs. US-born) instead of country of origin and have ensured that the reference category is provided for all dichotomous variables.

  • In the narrative, please add directionality and strength to the correlations. Add a short interpretation for age and marital status as both are negative and weakly correlated. Generally, if correlation less than 0.30 is weak.

AR: Thank you. We have added directionality and strength to the correlations in the narrative, and a brief interpretation of the age and marital status correlations (page 6).

  • Identifying which sociodemographics are relevant can guide future researchers in what to include in their analysis as covariates.

AR: Indeed. We have highlighted some relevant covariates that future studies can explore (page 6).

Discussion:

  • Provide a brief comparison between your two studies - Young vs. adults. Perhaps focus on acculturation and mean responses from the HCD scale which was very different across studies. Why could this be? This is more of a personal interest in understanding why there is a difference rather than answering your overall research question. However, this could also guide future use of the scale based on age and other indicators.

AR: Indeed. We have provided a brief discussion of the results for these two populations (page 8).

  • Is there any other study assessing the validity of the scale and its impact among Latinos? If so, please add it to the discussion to provide some context. See systematic review presented above.

AR: We have not identified another psychometric study of the 7-item scale to date (other than our own). The Bird and Lindly (2022) review mentioned just one other study with Latina/o/x populations: Sheppard et al. (2014) used a six-item scale but reported no psychometrics.

  • Factor loading seems similar in both studies. This could help researchers identify what other scales should be included, besides the HCD, in future research assessing healthcare discrimination.

AR: Indeed, these are important results, as the loadings remained similar despite the different Latino populations. We have added a sentence pointing out the need to supplement studies with other measures that can better capture “unequal treatment” such as medical mistrust (page 8).

  • I kept forgetting that they were churchgoers and just saw it as a convenience sample from a community setting. Please provide more justification on why it is important to consider churchgoers or not in healthcare discrimination. You mention this in the limitations but I think it is more of a discussion as I can see that there is some work already done comparing religious beliefs.

AR: We have expanded our statement about church-based samples, as there is evidence from other research pointing to increasing medical mistrust when parishioners are more closely engaged with their churches (see López-Cevallos et al, 2021; page 9).

  • Besides promoting the use of the scale among diverse Latino communities and

the need to test for validity, please provide a couple of implications for its use in the general population.

AR: Thank you. We have added a sentence at the end of the conclusion section with broader implications (page 9).

Limitations:

  • Provide a justification for small sample size in U.S. born Latino group plus

potential under or overestimation of model fit an overall factor analysis.

AR: Thank you. We have included a statement regarding the connection between the relatively small US-born sample size and CFA model fit (page 9).

  • On line 237: you mentioned that test-retest reliability was not done and that

experiences may be different across time. However, the scale assesses

discrimination EVER, not in a restricted period in time. I think the justification for this may be more logistical (e.g., surveying the same individuals again, end of the study) rather than experiencing differences. Please notice this and address properly. 

AR: Thank you. We have edited this statement to highlight the need for retesting the same individuals (test-retest reliability), rather than differences over time (page 9).

Reviewer 2 Report

Comments and Suggestions for Authors

This manuscript addresses an important measurement gap in the area of healthcare discrimination by focusing on a population that frequently experiences healthcare discrimination. The approach to assess the psychometric properties of the instrument designed to measure healthcare discrimination is appropriate. The following recommendations are offered to strengthen the manuscript's potential contributions to the field: 

  • The rationale provided to justify the manuscript's sample of Latino churchgoers is not compelling (i.e., "70% of Latino adults are religiously affiliated" as this rate is comparable to US population rates). Additionally, the study's sample only included Catholic Latinos. Were there other reasons for using the sample from the Sennet al., 2023 study?  Religious affiliation is quite a discrete sociodemographic variable which limits the study's generalizability. 
  • The use of the  Perceived Stress Scale to assess predictive validity did not yield significant results. It is recommended that the authors offer suggestions of alternative constructs to assess predictive validity in future studies.
  • In the discussion section, the authors recommend leveraging the HCD scale in relation to the complex and layered nature of discrimination in order to understand healthcare inequities. Can the authors provide more details about how such future research would look like?
  • The conclusions section seems underdeveloped as it does not include how the broader implications and importance of using the HCD scale to better understand the experiences of healthcare discrimination among Latinos  in light of their historically limited access to healthcare services in the United States. 

Author Response

Reviewer # 2

This manuscript addresses an important measurement gap in the area of healthcare discrimination by focusing on a population that frequently experiences healthcare discrimination. The approach to assess the psychometric properties of the instrument designed to measure healthcare discrimination is appropriate. The following recommendations are offered to strengthen the manuscript's potential contributions to the field: 

  • The rationale provided to justify the manuscript's sample of Latino churchgoers is not compelling (i.e., "70% of Latino adults are religiously affiliated" as this rate is comparable to US population rates). Additionally, the study's sample only included Catholic Latinos. Were there other reasons for using the sample from the Sennet al., 2023 study?  Religious affiliation is quite a discrete sociodemographic variable which limits the study's generalizability. 

AR: Thank you. Indeed, we mention as one of our study limitations that it included only Catholic Latinos (and the need for studies among Latinos of other religious affiliations; page 9). This is because the sample was drawn from the parent study (Derose et al., 2022, describes it in further detail -  not Senn et al., 2023).  

  • The use of the  Perceived Stress Scale to assess predictive validity did not yield significant results. It is recommended that the authors offer suggestions of alternative constructs to assess predictive validity in future studies.

AR: Indeed, we have edited our statement regarding the perceived stress scale variable: “perceived stress may have been too distal an outcome to use for testing predictive validity. The parent study did not include more proximal outcomes, such as the number of health care visits in the previous year, which should be explored in future studies” (pages 9-10).

  • In the discussion section, the authors recommend leveraging the HCD scale in relation to the complex and layered nature of discrimination in order to understand healthcare inequities. Can the authors provide more details about how such future research would look like?

AR: Thank you. We have added a statement suggesting what future research would entail.

  • The conclusions section seems underdeveloped as it does not include how the broader implications and importance of using the HCD scale to better understand the experiences of healthcare discrimination among Latinos  in light of their historically limited access to healthcare services in the United States (page 8). 

AR: Thank you. We have expanded our conclusions to address the broader implications of using the HCD scale (page 9).